# Arteria Praebronchialis (AP) Found on MDCT: An Updated Incidence and Branching Patterns

**DOI:** 10.3390/diagnostics13172744

**Published:** 2023-08-24

**Authors:** Bo Mi Gil, Kyongmin Sarah Beck, Kyung Soo Kim, Dae Hee Han

**Affiliations:** 1Department of Radiology, Bucheon St. Mary’s Hospital, College of Medicine, The Catholic University of Korea, 327, Sosa-ro, Wonmi-gu, Bucheon-si 14647, Gyeonggi-do, Republic of Korea; xhxhh@catholic.ac.kr; 2Department of Radiology, Seoul St. Mary’s Hospital, College of Medicine, The Catholic University of Korea, 222, Banpo-daero, Seocho-gu, Seoul 06591, Republic of Korea; lepolder@gmail.com; 3Department of Thoracic and Cardiovascular Surgery, Seoul St. Mary’s Hospital, College of Medicine, The Catholic University of Korea, 222, Banpo-daero, Seocho-gu, Seoul 06591, Republic of Korea; cskks@catholic.ac.kr

**Keywords:** arteria praebronchialis, anatomic variation, vascular anomaly, pulmonary artery variant

## Abstract

Preoperative detection of the arteria praebronchialis (AP), a rare variant mediastinal branch of the left pulmonary artery, can be crucial to a successful left-lung surgery; if the AP is overlooked and ligated during surgery, the blood supply to the remaining lobe may be compromised. The purpose of this study was to update the incidence and branching patterns of the AP. From 18 April 2012 to 31 December 2022, contrast-enhanced CT was screened by one radiologist for the presence of AP. Branching patterns of the AP were analyzed by three thoracic radiologists. The incidence of AP was updated to 0.068% (18/26,310) from the previously reported 0.03%; the incidence of AP for male and female patients was 0.110% and 0.017%, respectively. AP supplied only the LLL in 10 cases and both the lingular division of LUL and LLL in nine cases. Dual segmental supply by both the AP and the normal left descending pulmonary artery existed in 15 cases; exclusive segmental supply by either artery existed in four cases. The AP supplies either the LLL alone or both LLL and the lingular division of LUL, and its incidence is not negligible in the male population, necessitating routine surveillance prior to pulmonary resection.

## 1. Introduction

Preoperative detection of pulmonary vascular variations can be crucial to successful lung surgery. One particular variant requires careful attention during preoperative evaluation: the arteria praebronchialis (AP). The AP is an aberrant branch of the left pulmonary artery that arises either simultaneously or prior to the left descending pulmonary artery (LDPA), crosses in front of the left mainstem bronchus (hence the name “praebronchial”) and descends in an inferior direction along the mediastinal margin (Figure 1). If this artery is overlooked and ligated during a left-lung surgery, the blood supply to the remaining lobe may be compromised; therefore, radiologists should be able to recognize the AP and describe its course before the surgery.

The AP was first reported by an anatomist Buntaro Adachi in 1928, long before the invention of CT [1]. The second case was reported in 1985, which was also the first case that saw this variant being imaged (catheter angiography) in a live patient. The first CT case was reported in 1995 [2]. In the era of MDCT, multiple sporadic reports have been made in the following years under many different names, including “pars medialis” [3] and “abnormal branching of the left lingular pulmonary artery” [4], among others. The anomalous arteries were all AP in fact, but because it has been reported under various names, this variant is not readily recognized and often confused with other variants by radiologists or surgeons.

In 2015, one group of authors in our institution analyzed the branching pattern of the AP in four patients, along with its estimated incidence [5]. Since then, 8 years have passed and more cases of the AP have been collected; with more cases to analyze, the incidence and branching patterns of the AP appeared to be different from the original report. Knowing the accurate incidence and branching patterns of the AP would be helpful in the preoperative planning of left lung surgery. Hence, this study aims to provide an updated incidence and branching patterns of the AP based on a larger patient cohort compared to previous studies.

## 2. Materials and Methods

### 2.1. Study Population and Incidence

Since the first case of the AP was incidentally discovered by one thoracic radiologist (D.H.H., 20 years of experience in thoracic radiology) on 18 April 2012, all consecutive contrast-enhanced CTs (CECTs) interpreted by the same radiologist were screened for the presence of AP until 31 December 2022, regardless of the indications for undergoing CT. Two other thoracic radiologists (B.M.G. and K.S.B., 6 years and 8 years of experience in thoracic radiology, respectively) reviewed the CTs and confirmed the presence of the AP. The incidence of the AP was calculated by dividing the number of patients with the AP found after the initial accidental discovery by the number of chest multidetector computed tomography (MDCT) with enhancement interpreted by D.H.H. after the initial accidental discovery of the AP on 18 April 2012 until 31 December 2022. This study population includes four patients reported in a previous study [5].

### 2.2. CT Technique

Chest MDCT with enhancement was performed on scanners (SOMATOM Definition Edge and SOMATOM Force, Siemens Healthineers, Erlangen, Germany) using a slice thickness of 1 mm reconstructed at 0.75 mm intervals with contrast injection by a mechanical injector at a rate of 3 mL/s, for a total dose of 100 mL. Axial, sagittal and coronal images were reconstructed at 3 mm intervals and were transferred to a PACS workstation (TaeYoung Soft, Seoul, Republic of Korea).

### 2.3. Image Interpretation

The course, branching pattern, segmental supply, and diameter of the AP were analyzed by three thoracic radiologists. The size of the AP was recorded in comparison with the normal left descending pulmonary artery (LDPA), which is also known as the interlobar artery. Additionally, the presence of other coexisting variations, including the right pulmonary artery, vein, the heart and major airways, was evaluated [6]. Conclusions were reached for each subject by consensus.

## 3. Results

### 3.1. Incidence

Out of a total of 26,310 patients (14,553 male) who underwent CECT scans, 18 patients with AP were identified, yielding an incidence rate of 0.068%. Of 18 patients, 16 (88.9%) were male, and the calculated incidence of AP for male and female patients were 0.110%. and 0.017%, respectively.

### 3.2. Lobar and Segmental Supply Patternf of the AP

The course of AP was analyzed in a total of 19 patients (first patient and subsequent 18 patients), and a detailed segmental supply of each patient is presented in Table 1. AP supplied the left lower lobe (LLL) exclusively (*n* = 10; 52.6%) or both the LLL and the lingular division of the left upper lobe (LUL) (*n* = 9; 47.4%). In no cases did the AP supply the superior division (S1+2, S3) of the LUL or the superior segment (S6) of the LLL. The AP provided blood supply to S7+8 in 18 of 19 cases (94.7%). The AP and the LDPA supplied one or more segments together in 15 of 19 cases (78.9%); mutually exclusive segmental blood supply of the AP and the LDPA was seen in 4 of 19 cases (21.1%). These results are summarized in Table 2.

### 3.3. Diameter of the AP

Among the 19 patients, the diameter of the AP was smaller than, greater than, or equal to the LDPA in 10, 4, and 5 patients, respectively. AP that is smaller than the LDPA either supplied 1 segment (*n* = 2), 2 segments (*n* = 6), or 3 segments (*n* = 2). AP that is of an equal size to the LDPA either supplied two segments (*n* = 1), 3 segments (*n* = 3), or five segments (*n* = 1). AP that is larger than LDPA supplied either two segments (*n* = 3) or three segments (*n* = 1). The results are summarized in Table 3.

### 3.4. Presence of Other Accompanying Variations

No other variations of the heart, the right pulmonary vein, or major airways were observed in the study patients. However, an accompanying contralateral variation, aberrant right A7, was found in three (15.8%) patients (Figure 2).

## 4. Discussion

Our study updates the incidence of the AP from 0.03% [5] to 0.065%, with the largest number of AP cases collected. Until recently, the AP was known to be a very rare variant, probably because the presence and course of the AP can only be accurately evaluated with MDCT. With the increased number of MDCT exams, the incidence of the AP may also gradually increase, as proven by the updated incidence of 0.065% from 0.03% in our study; this rise in incidence may be due to the overall increased number of cumulative CT exams that were screened for the presence of the AP. As the number of MDCT exams increases along with the number of detection and surgery of early-stage lung cancer, recognition of this anomaly may become more crucial, from a surgical point of view.

Failure to detect AP in pre-surgical imaging studies can be detrimental to the safety of a left-lung surgery; if the AP that supplies both the lingular division and the LLL, as in about half of our study patients, is overlooked and ligated during a left-lung surgery, the blood supply to the remaining lobe may be compromised. A greater problem would arise if the segmental supply of such AP was mutually exclusive with the LDPA, as in 21.1% of our study population. The majority of recent thoracic surgeries are performed using video-assisted thoracoscopic surgery (VATS) and securing an adequate surgical field of view in VATS is more challenging compared to open thoracotomy, because it is more difficult to see the whole lung in VATS; anomalous vessels may be overlooked in such situations, highlighting the importance of preoperative detection of vascular anomalies. Performing an anatomical pulmonary resection of the left lung without the preoperative conception of AP would be more difficult and take a longer time, compared to cases without arterial variations. Therefore, preoperative detection and accurate description of the branching pattern and course of the AP, in addition to comparing and confirming the actual branching pattern and course during surgery, seem crucial to the success of a left-lung surgery.

In this study, the calculated incidence of the AP for male and female patients was 0.110% and 0.017%, respectively, demonstrating a strong male predominance, in line with previous reports [2,3,4,5,6,7,8,9,10,11,12,13,14,15,16,17]. As the AP is found in approximately 1 in 1000 males, a routine preoperative CT examination for the presence of the AP before any left-lung surgery seems particularly important in male patients. In addition, the AP has been reported only among eastern Asians to date, possibly suggesting a racial predilection, which remains to be confirmed in the future.

After being first reported by Buntaro Adachi in 1928 during the observation of vascular anatomy using corrosion preparation technique [1], the AP has been reported under various names [2,3,4,6,7,8,9,10,11,12,13,14,15,16,17]: “abnormal branching of left pulmonary artery”, “aberrant ramus of left pulmonary artery”, “(dangerous) mediastinal basal pulmonary artery”, “extremely rare branching pattern of the left pulmonary artery”, “variant of the left posterior basal segmental pulmonary artery”, “abnormal branching of the left anterobasal pulmonary artery”, “abnormal branching of left basal pulmonary artery from left main pulmonary artery”, “abnormal branching of the left lingular pulmonary artery”, “mediastinal basal pulmonary artery”, “common trunk from the left main pulmonary artery”, “mediastinal inferior lobar branch” and “pars medialis”. Because they were reported under different names, recognizing them as the same entity is extremely difficult, adding confusion when interpreting CT exams. Adopting a unified terminology by using the initially proposed name, AP, may be necessary to standardize the nomenclature and reduce confusion.

Our study has revealed that the AP is an independently branching variation, distinct from the mediastinal-type lingular artery. Some have considered the AP as a variant of the mediastinal type of lingular artery [4,9,10,12,18,19]. However, in our study, the AP was found to supply segments A7+8 in 94.74% of the cases, and about 26% of the AP cases did not have a lingular branch. Consequently, it would be difficult to consider the AP as a variant of the mediastinal type of lingular artery.

In this study, the relationship between the A3 branch and the AP has been reassessed through discussions among three thoracic radiologists, and the branching patterns of three cases from the previous report [5] have been updated. The A3 branch, which was initially considered to course from the AP to the superior division of LUL, is now considered to originate either simultaneously with AP from the left pulmonary artery or just prior to the branching of AP, rather than originating from the AP (Figure 3). In other words, A3 is considered to arise from the LDPA, not from the AP.

One study investigating the branching pattern of the most common variation, the mediastinal type of lingular artery, demonstrated that the volume of blood supply to the lingular division varied depending on its diameter [20]. Based on this observation, it was hypothesized that AP with a larger diameter may potentially branch into more segments, serving as a substitute for the LDPA. However, our results showed no correlation between the diameter of the AP and the number of segments being supplied. Among ten patients with a smaller diameter of the AP compared to LDPA, the AP supplied one segment only in two cases. The AP supplying five segments (S4, S5, S7+8, S0, S10) had an equal diameter with the LDPA. AP supplying two or three segments demonstrated varying diameters (smaller, greater, or equal to) compared to LDPA.

In the previous study [5], 50% of the AP was reported to be associated with the contralateral A7 branch variant, but in our study, only about 16% of the AP was associated with the contralateral A7 branch variant. The presence of a contralateral variant may have been incidental and less associated than previously thought. However, despite this, the co-occurrence of rare cases suggests that these two variants are not entirely independent but rather have a tendency to occur concurrently, indicating an associated condition.

Identifying the branching patterns of the AP was an extremely elaborate process. One example, the A5, which is very thin and could be easily missed, required careful observation using 0.75 mm thin slice images for precise confirmation (Figure 4). Systematic approaches to identifying and tracing the AP have not been described in previous reports. Since tracing the AP may be challenging for less experienced radiologists, we aim to present the following algorithm for identifying, tracking, and describing the AP (Figure 5). First, when a mediastinal-type variant branch (so-called pars mediastinalis) of the left pulmonary artery is encountered (especially in the sagittal plane), one should trace the branch to see if this supplies only the lingular division (so-called mediastinal-type of lingular artery) or also the LLL (the AP). Second, it is necessary to examine whether AP branches into both the LUL and LLL or if it only branches into the LLL. Third, the relationship between the branches of the AP and the target segment to be resected needs to be investigated. Lastly, extra attention should be paid during vascular anastomosis by rechecking whether the target segment has a dual blood supply or not.

There are some limitations in our study. First, the results were not surgically or angiographically confirmed. However, the MDCT findings of the AP were straightforward, can be detected with prior knowledge if not overlooked, and surgical correlation was confirmed in multiple other cases in the literature. Second, since we did not have vessel-tracking software, only visual assessment by thoracic radiologists was conducted. The identification of each branching pattern was extremely elaborate, but careful observation by three thoracic radiologists was done using thin slice images, which we believe could guarantee an accurate assessment of the vessels.

## 5. Conclusions

In conclusion, the AP supplies either the LLL alone or both LLL and the lingular division of LUL, and its incidence (0.110%) is not negligible in the male population, necessitating routine surveillance with MDCT prior to a left-lung surgery. Radiologists should be aware of this variant and its common branching patterns.

## Figures and Tables

**Figure 1 diagnostics-13-02744-f001:**
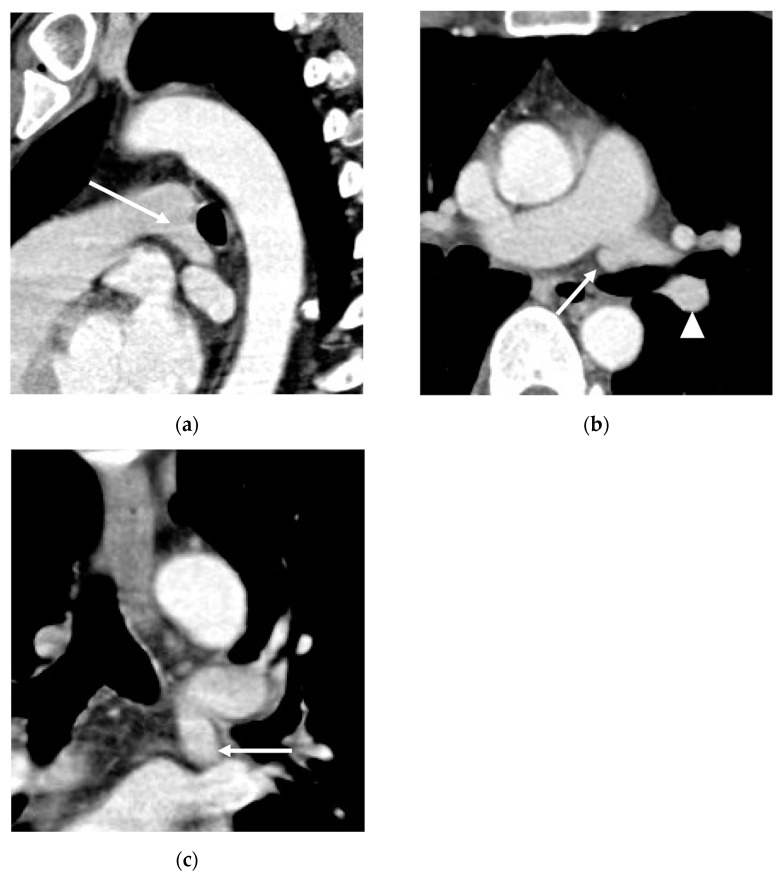
Axial (**a**), coronal (**b**) and sagittal (**c**) CT images of a 78-year-old male patient show the course of AP (arrows) originating from the medial margin of the left pulmonary artery, running anteriorly to the left main stem bronchus. The normal left descending pulmonary artery (LDPA; arrowheads) branches after the AP.

**Figure 2 diagnostics-13-02744-f002:**
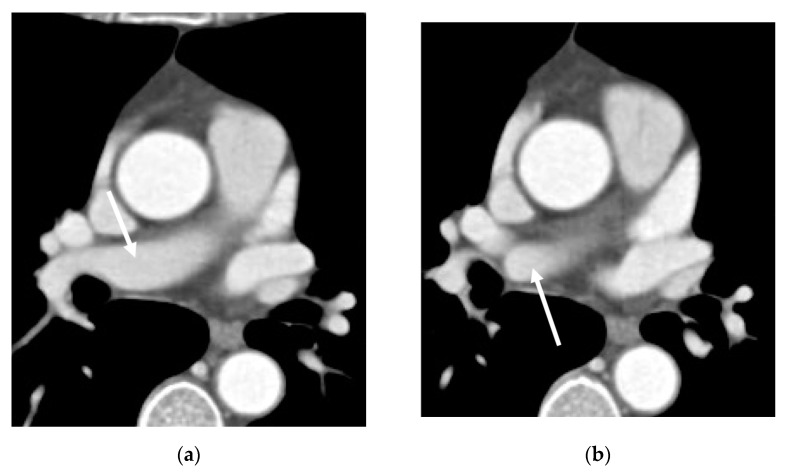
Axial CT images (**a**–**c**) of a 58-year-old female show aberrant A7 branching from the right pulmonary artery (arrows) before branching into the right normal descending pulmonary artery.

**Figure 3 diagnostics-13-02744-f003:**
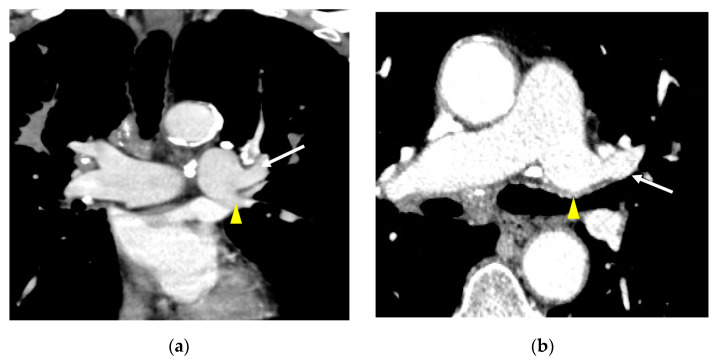
Assessment of A3 branching site. Axial CT images of a 66-year-old male patient (**a**,**b**) and a 29-year-old male patient (**c**,**d**) show AP (arrows) and A3 (arrowheads) branching simultaneously.

**Figure 4 diagnostics-13-02744-f004:**
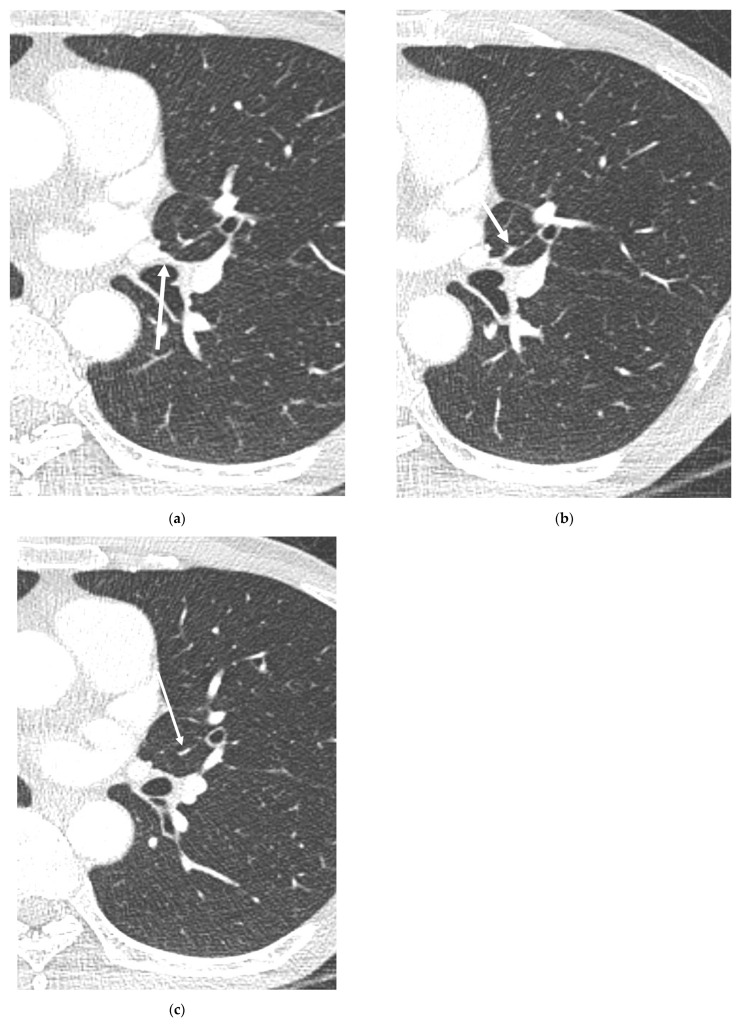
A5 origin that is easy to miss. Axial CT images (**a**–**c**) of a 67-year-old female patient show a very thin A5 branch (arrow), which required careful observation using 0.75 mm thin slice images for precise confirmation.

**Figure 5 diagnostics-13-02744-f005:**
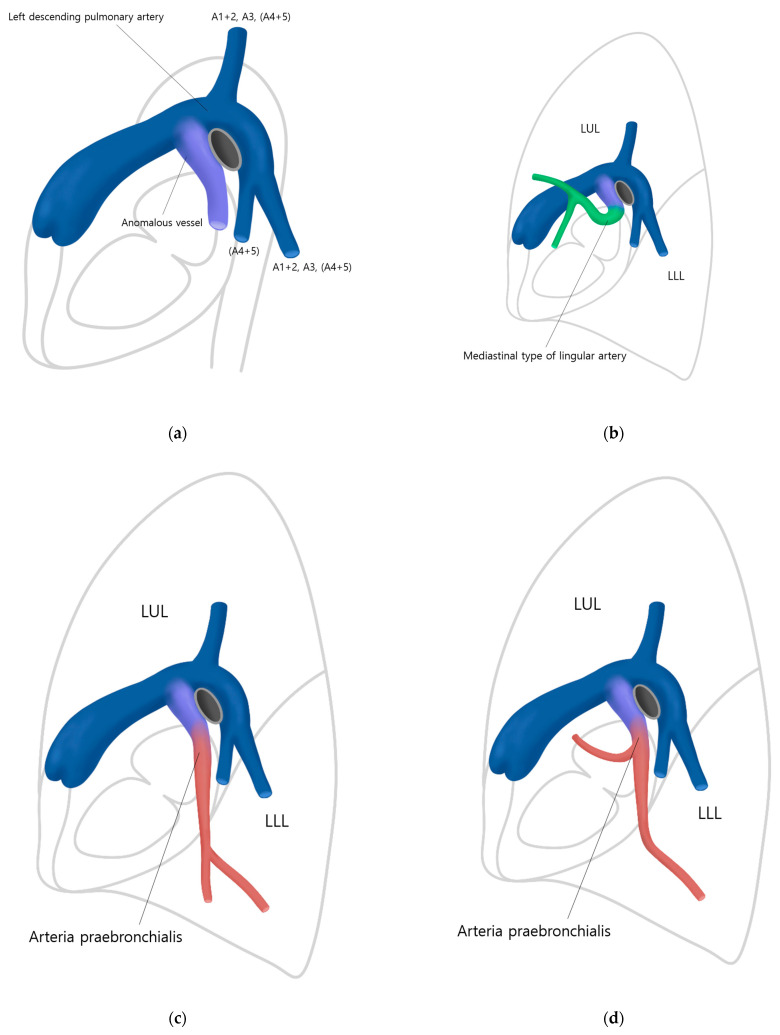
An algorithm for identifying, tracking, and describing the AP. (**a**) First, identify the anomalous vessel branching from the anterior aspect of the left pulmonary artery on a sagittal image (appearing as a laterally inclined Y shape). Second, it is necessary to differentiate whether the subsequent anomalous branching vessel is a mediastinal type of lingular artery or the AP. (**b**) In the case of the mediastinal type of lingular artery, the vessel branches towards only the left upper lobe (LUL). (**c**,**d**) In the case of the AP, it descends vertically anterior to the left lower lobe (LLL), and it may supply only the LLL (**c**) or both LUL and the LLL (**d**). (**e**,**f**) The course of the AP would be like this on axial images and a 3D reformatted CT image. (blue = left main pulmonary artery and left descending pulmonary artery. purple = anomalous vessel. green = mediastinal type of lingular artery. red = arteria praebronchialis).

**Table 1 diagnostics-13-02744-t001:** Detailed Segmental Supply of AP and LDPA in Each Patient.

Patient	Age	Sex	Superior Division of LUL		Lingular Division of LUL		LLL				Number of Segments Supplied by AP		
			S1+2	S3	S4	S5	S6	S7+8	S9	S10	Lingular Division	LLL	Total
Patient 1	29	M	X	X	L	L	L	A	L	B	0	2	2
Patient 2	66	M	X	X	L	A	L	A	A	L	1	2	3
Patient 3	72	M	X	X	L	L	L	A	L	L	0	1	1
Patient 4	64	M	X	X	B	A	L	B	B	B	1	4	5
Patient 5	59	M	X	X	L	A	L	B	L	L	1	1	2
Patient 6	62	M	X	X	L	L	L	B	L	B	0	2	2
Patient 7	53	F	X	X	L	A	L	B	L	B	1	2	3
Patient 8	51	M	X	X	L	L	L	B	L	A	0	2	2
Patient 9	63	M	X	X	L	B	L	L	L	B	1	1	2
Patient 10	69	F	X	X	L	A	L	A	L	L	1	1	2
Patient 11	76	M	X	X	L	B	L	B	L	B	1	2	3
Patient 12	81	M	X	X	L	L	L	B	L	L	0	1	1
Patient 13	78	M	X	X	L	L	L	B	L	B	0	2	2
Patient 14	41	F	X	X	L	A	L	B	L	B	1	2	3
Patient 15	34	M	X	X	L	L	L	B	B	B	0	3	3
Patient 16	72	M	X	X	L	L	L	A	B	B	0	3	3
Patient 17	65	M	X	X	L	A	L	A	L	L	1	1	2
Patient 18	72	M	X	X	L	L	L	B	L	B	0	2	2
Patient 19	46	M	X	X	L	L	L	B	L	B	0	2	2

A = supplied by the AP, B = supplied by both AP and LDPA, AP = arteria praebronchialis, F = female, L = supplied by the LDPA, LDPA = left descending pulmonary artery, M = male, S = segment, X = supplied by neither AP nor LDPA.

**Table 2 diagnostics-13-02744-t002:** Segmental Supply of AP and LDPA in 19 patients.

Supplying Artery	Segments							
	LUL Superior Division		LUL Lingular Division		LLL			
	S1+2	S3	S4	S5	S6	S7+8	S9	S10
AP	0	0	1	9	0	18	4	13
LDPA	0	0	19	12	19	13	18	18
Both AP and LDPA	0	0	1	2	0	12	3	12
Exclusively by AP	0	0	0	7	0	6	1	1
Exclusively by LDPA	0	0	18	10	19	1	15	6

AP = arteria praebronchialis, LDPA = left descending pulmonary artery, S = segment.

**Table 3 diagnostics-13-02744-t003:** The relationship between size of the AP and the number of segments supplied by AP.

	The Size of AP, Compared with LDPA	Number of Segments Supplied by AP
Patient 1	Smaller	2
Patient 2	Larger	3
Patient 3	Smaller	1
Patient 4	Equal	5
Patient 5	Equal	2
Patient 6	Smaller	2
Patient 7	Equal	3
Patient 8	Larger	2
Patient 9	Smaller	2
Patient 10	Larger	2
Patient 11	Equal	3
Patient 12	Smaller	1
Patient 13	Larger	2
Patient 14	Smaller	3
Patient 15	Smaller	3
Patient 16	Equal	3
Patient 17	Smaller	2
Patient 18	Smaller	2
Patient 19	Smaller	2

AP = arteria praebronchiolis, LDPA = left descending pulmonary artery.

## Data Availability

The data presented in this study are available on request from the corresponding author.

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
