# Peer review of "Arteria Praebronchialis (AP) Found on MDCT: An Updated Incidence and Branching Patterns"

_diagnostics, 2023, doi:10.3390/diagnostics13172744_

Round 1

Reviewer 1 Report

I should congratulate authors for this paper. I would like to get some more clarifications.

1) What was the population studied here? Did it include all patients who underwent MDCT or those undergoing Lung surgery?

2) What was the percentage of difference of opinion of 3 radiologists in confirming diagnosis of AP ? How many cases were diagnosed by all 3 of them?

3) How many patients undergo Angiography/ surgery among the patients? Did they find any change in diagnosis compared to the finding in MDCT?

4) Do you recommend MDCT before all planned Left lung surgery by VATS ?

I hope these clarifications will make your manuscript more scientific.

Author Response

Comments and Suggestions for Authors from Reviewer 1

Modified parts are indicated in the revised manuscript with memos.

 1) What was the population studied here? Did it include all patients who underwent MDCT or those undergoing Lung surgery?

Thank you for this valuable comment. The study population consisted of all patients undergoing MDCT for various reasons, not just those undergoing lung surgery. We have clarified this by modifying a sentence in the materials and methods as follows:

“Since the first case of the AP was incidentally discovered by one thoracic radiologist (D.H.H., 20 years of experience in thoracic radiology) on April 18, 2012, all consecutive contrast-enhanced CTs (CECTs) interpreted by the same radiologist was screened for the presence of AP until December 31, 2022, regardless of the indications for undergoing CT.” (Reviewer 1, comment 1).

2) What was the percentage of difference of opinion of 3 radiologists in confirming diagnosis of AP? How many cases were diagnosed by all 3 of them?

Thank you for this insightful comment. The presence of AP was screened and detected by one thoracic radiologist, as described in the materials and methods section (“Since the first case of the AP was incidentally discovered by one thoracic radiologist (D.H.H., 20 years of experience in thoracic radiology) on April 18, 2012, all consecutive contrast-enhanced CTs (CECTs) interpreted by the same radiologist was screened for the presence of AP until December 31, 2022, regardless of the indications for undergoing CT.” in page 3 of the revised manuscript). Without prior knowledge, the AP may be overlooked, but with the appropriate knowledge, the detection and diagnosis are straightforward; therefore, there wasn’t any different opinions from two other radiologists regarding the diagnosis of the AP. They all agreed with and confirmed the diagnosis of the AP in 19 patients in the analysis. To clarify, following sentence was added to page 3 of Materials and Methods section of the revised manuscript:

“Two other thoracic radiologists (B.M.G. and K.S.B., 6 years and 8 years of experience in thoracic radiology, respectively) reviewed the CT and confirmed the presence of the AP.” (Reveiwer 1, comment 2).

3) How many patients undergo Angiography/ surgery among the patients? Did they find any change in diagnosis compared to the finding in MDCT?

Thank you for this valuable comment. None of the 19 patients underwent angiography or lung surgery, so we were unable to confirm the presence of the AP surgically or angiographically. However, as we have explained in the limitations of the discussion section (page 7), the detection and diagnosis of the AP on MDCT were straightforward, and surgical correlation was confirmed in multiple other cases in the literature. We have modified a sentence in the limitations of the discussion section in the revised manuscript to clearly explain the concerns raised by Reviewer 1:

“First, the results were not surgically or angiographically confirmed. However, the MDCT findings of the AP were straightforward, can be detected with prior knowledge if not overlooked, and surgical correlation was con-firmed in multiple other cases in the literature.” (Reviewer 1, comment 3).

4) Do you recommend MDCT before all planned Left lung surgery by VATS ?

 Thank you for this insightful comment. Yes, I think it should be. We have slightly modified the conclusion to clearly emphasize this:

“In conclusion, the AP supplies either the LLL alone or both LLL and the lingular division of LUL, and its incidence (0.110%) is not negligible in the male population, necessitating routine surveillance with MDCT prior to a left-lung surgery.” (Reviewer 1, comment 4).

Reviewer 2 Report

Thank you to the authors for sharing a such experience: it is useful to know the presence of praebronchialis artery to reduce the risk of its ligature.

Just two comments:

1) avoid to start the sentences with numbers (in results not to start with 18)

2) please could you provide some radiological reconstructions? Drawings are fantastic but a reconstruction is useful

Minor revisions

Author Response

Comments and Suggestions for Authors from Reviewer 2

Modified parts are indicated in the revised manuscript with memos.

1) Avoid to start the sentences with numbers (in results not to start with 18)

Thank you for the comment. We have changed the sentence in the results section of the revised manuscript accordingly (Reviewer 2, comment 1).

2) Please could you provide some radiological reconstructions? Drawings are fantastic but a reconstruction is useful

Thank you for this insightful comment. We have added Figure 1F, a 3D reformatted CT image, as you have suggested (Reviewer 2, comment 2).